# Call combinations and compositional processing in wild chimpanzees

Maël Leroux [1,2,3] ✉, Anne M. Schel[4], Claudia Wilke[1,2,3], Bosco Chandia[2], Klaus Zuberbühler[2,3,5,6], Katie E. Slocombe[7] & Simon W. Townsend [1,3,8]

Through syntax, i.e., the combination of words into larger phrases, language can express a limitless number of messages. Data in great apes, our closest-living relatives, are central to the reconstruction of syntax's phylogenetic origins, yet are currently lacking. Here, we provide evidence for syntactic-like structuring in chimpanzee communication. Chimpanzees produce "*alarm-huus*" when surprised and "*waa-barks*" when potentially recruiting conspecifics during aggression or hunting. Anecdotal data suggested chimpanzees combine these calls specifically when encountering snakes. Using snake presentations, we confirm call combinations are produced when individuals encounter snakes and find that more individuals join the caller after hearing the combination. To test the meaning-bearing nature of the call combination, we use playbacks of artificially-constructed call combinations and both independent calls. Chimpanzees react most strongly to call combinations, showing longer looking responses, compared with both independent calls. We propose the "*alarm-huu + waa-bark*" represents a compositional syntactic-like structure, where the meaning of the call combination is derived from the meaning of its parts. Our work suggests that compositional structures may not have evolved de novo in the human lineage, but that the cognitive building-blocks facilitating syntax may have been present in our last common ancestor with chimpanzees.

Human language is a highly productive communication system whereby new meaning can be created syntactically through the combination of existing meaning-bearing units (or words)[1]. Syntax can take different forms that can be differentiated according to the semantic relationship between the combination and the comprising units. Combinatorial syntax[2], for example, includes structures where the meaning generated is independent from the meaning of the parts (e.g., idioms, "*cry wolf*")[2,3]. Compositional syntax[2], on the other hand, designates structures where the meaning of the whole is directly derived from the meaning of the parts (e.g. "*careful with the wolf*")[2,3].

Compositional syntax can be further decomposed into various, more specific, configurations such as predication (e.g., "*the wolf howled*"), modification (e.g., "*black wolf*"), or simple coordination (i.e. not involving any dependencies, e.g. "*wolf and dog*")[3] (see ref. 3 for further discussion). The communicative importance of syntax and its role in the infinite generative power of language is uncontroversial[4]. More contentious debate surrounds its evolutionary origins, specifically whether this trait is truly unique to our species' communication system[3,5–7]. An emerging body of observational and experimental data has highlighted similar abilities in the primate lineage[8–11], suggesting

[1]Department of Comparative Language Science, University of Zürich, Zürich, Switzerland. [2]Budongo Conservation Field Station, Masindi, Uganda. [3]Center for the Interdisciplinary Study of Language Evolution (ISLE), University of Zürich, Zürich, Switzerland. [4]Animal Behaviour and Cognition, Utrecht University, Utrecht, Netherlands. [5]Department of Comparative Cognition, Institute of Biology, University of Neuchâtel, Neuchâtel, Switzerland. [6]School of Psychology and Neuroscience, University of St Andrews, St Andrews Scotland, UK. [7]Department of Psychology, University of York, York, UK. [8]Department of Psychology, University of Warwick, Coventry, UK. ✉e-mail: maelmaodez.leroux@gmail.com

the rudimentary capacity to sequence meaning-bearing vocal units together, the core foundations of syntax[3] (but see ref. [12,13]), could have emerged as early as in our last common ancestor with catarrhines and platyrrhines around 45 million years ago[14]. For example, putty-nosed monkeys combine two distinct calls referring to predators or disturbances into a larger sequence that is produced in the context of group movement. In this instance, the meaning of the combination (/move/) is unrelated to the meaning of the comprising parts (/eagle/; /disturbance/) and therefore represents a case of combinatorial syntactic-like structuring in this species[15]. Furthermore, Campbell's monkeys affix an "-oo" element onto the end of terrestrial and aerial alarm calls which serves to modify the meaning of the alarm calls in a predictable way such that they designate a less specific disturbance but in the same physical space (i.e., on the ground or in the canopy)[8,16]. Since the meaning of the combination appears to be directly related to the meaning of the parts (/threat-type/ + /low-urgency/), this example has been repeatedly interpreted as a rudimentary compositional syntactic-like structure[8,17,18]. However, equivalent data in more distantly-related species indicate syntax can evolve through convergent evolution, ultimately complicating an evolutionary ancient account of syntax in the primate lineage[19–23]. For instance, both pied babblers and Japanese tits have been shown to combine an alarm call with a recruitment call when encountering a threat that requires recruitment[22,23]. These structures have also been argued to represent rudimentary forms of compositional syntactic-like constructions where the meaning of the whole is related to the meaning of the comprising units (/threat/ + /come here/) akin to what has been termed *coordination* in human language[3,22,23]. To disentangle whether syntactic structuring of signal combinations is a convergent or homologous trait within the primate lineage, comparable data in great apes are central. Several observational studies have already documented combinations of meaning-bearing units into larger structures in all four great ape species: orangutans[24], gorillas[25,26], bonobos[27–30] and chimpanzees[31–33], suggesting the potential for syntactic-like structuring in this clade. Whilst promising, to date, no systematic experimental work, key to demonstrating the syntactic-like nature of these structures, has been conducted. In this study, we aimed to bridge this gap by experimentally investigating whether wild chimpanzees produce meaningful combinations of specific call types.

Chimpanzees produce "alarm-huus" (AH) when they are frightened or surprised (e.g., earth tremors, snakes, dead monkeys, researcher's waterproof cloaks[34,35]). Another important call type in this species is the "waa-bark" (WB) which is produced in a range of social and ecological contexts such as hunting, predator encounters, inter-community encounters and aggression and has been argued to play a role in recruiting individuals to the caller[34,36–38]. Critically, previous work has shown that chimpanzees combine these two calls into the "alarm-huu + waa bark" structure (AH-WB) at frequencies higher than expected by chance[33] and additional natural observations, though rare, suggest chimpanzees do this when encountering a snake, specifically when isolated from other individuals but still within earshot ($N = 2$ observations in 18 months, see Supplementary Note 1 for more details and Fig. S1 for spectrograms of the calls). We hypothesized this call combination might therefore function as a recruitment signal, particularly in dangerous situations, and potentially represent a compositional syntactic-like structure, with the meaning of the whole (*recruitment to a threat*) being a product of the meaning of its parts (*threat + recruitment*) (see ref. [22,23] for similar combinations in birds).

We tested this hypothesis by presenting chimpanzees with model snakes (see Methods) and examining call combination production. Following presentations, we predicted more individuals in the audience would join the caller after the production of the call combination compared to when no combinations are produced (see methods for definition of joining individual). We also probed the function of the "alarm-huu + waa-bark" combination and its syntactic-like nature by

conducting playback experiments, the gold standard for investigating meaning attribution in animal vocalizations[39]. Specifically, we evaluated the responses of individuals to both singly-occurring calls (AH and WB) to characterize their typical response to each call type alone, and then compared these to the response to an artificial "alarm-huu + waa-bark" combination (see Methods for more details). Here, we predicted weak reactions from recipients to "alarm-huus" (i.e., long latency and short time looking at the loudspeaker) since they convey information regarding unspecific, non-urgent threats[22]. We predicted stronger responses to the "waa-barks" (i.e., shorter latency but short time looking at the loudspeaker) as it is a more abrupt intense sounding call that is likely to attract the attention of the recipient quickly, but may not evoke a long orientation response as the reason for recruitment is unclear[22]. Finally, we predicted the strongest responses to the call combination (short latency and long time looking at the loudspeaker) since a threat requiring recruitment is more of an urgent situation than recruitment alone ("waa-bark") or warning of a potential threat ("alarm-huu").

## Results

### Presentation experiments

In line with naturalistic observations, snake presentations elicited "alarm-huu + waa-bark" combinations. Specifically, in 9 out of 21 trials (43%), an "alarm-huu + waa-bark" sequence was produced by the subject within the first two minutes after discovering the snake. In the remaining trials (12/21; 57%), "alarm-huus" were consistently produced, but never "waa-barks".

We investigated whether the production of "alarm-huu + waa-bark" call combinations influenced the number of individuals joining the caller. Individuals joined the subject in 47% of trials (10/21), 90% (9/10) of which were accompanied by the production of a call combination by the subject. A GLMM with a Poisson family showed significantly more individuals joined subjects after the production of a call combination (GLMM$_{individuals-recruited}$: ß ± SE = 3.7 ± 0.771; z = 4.798; $P < 0.001$, see Fig. 1). Together, these results suggest that chimpanzees appear to produce the "alarm-huu + waa-bark" call combination when encountering snakes and its production is associated with the recruitment of conspecifics.

However, there are a number of issues that these data in isolation cannot adequately address. First, as we did not observe any snake

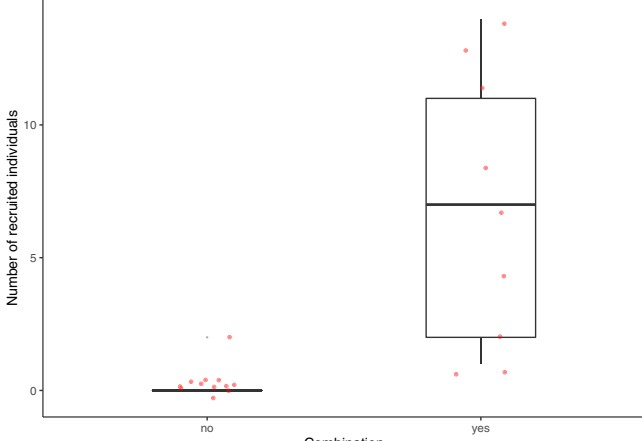

**Fig. 1 | Number of recruited individuals according to the production of the call combination.** yes: the subject produced an "alarm-huu + waa-bark" combination upon the discovery of the snake; no: the subject did not produce an "alarm-huu + waa-bark" combination upon the discovery of the snake. Red dots show the raw data. The boxes display the median value and 25 and 75% quartiles; the whiskers are extended to the most extreme value inside the 1.5-fold interquartile range. Trials = 21, individuals = 13. Source data are provided as a Source Data file.

**Table 1 | Post hoc analyses indicating the influence of play-back condition on the looking duration and the latency to look towards the loudspeaker**

| GLMM$_{looking-duration}$ | Estimate | SE | z | P |
|---|---|---|---|---|
| AH / WB | 0.37 | 0.469 | 0.788 | 0.430585 |
| AH / AH-WB | 1.756 | 0.453 | 3.856 | **0.000115** |
| WB / AH-WB | 1.376 | 0.442 | 3.116 | **0.00183** |
| GLMM$_{latency-to-look}$ | Estimate | SE | Z | P |
| AH / WB | −3.318 | 0.606 | −5.474 | **1.42e-05** |
| AH / AH-WB | −2.815 | 0.672 | −4.341 | **4.42e-08** |
| WB / AH-WB | 0.403 | 0.646 | 0.624 | 0.533 |

*AH* Singly-occurring "*alarm-huu*", *WB* Singly-occurring "*waa-bark*", *AH-WB* "*alarm-huu + waa-bark*" combination. GLMM$_{looking-duration}$: $\chi^2 = 17.259$, Df = 2, $P < 0.001$; GLMM$_{latency-to-look}$: $\chi^2 = 33.535$, Df = 2, $P < 0.001$ respectively, *SE* Standard Error
Bold indicate statistical significance.

model encounters that elicited only "*waa-barks*" (i.e. without "*alarm-huus*"), we cannot disentangle whether the recruitment of individuals was a response to the call combination, or merely to the presence of the "*waa-bark*" call. Secondly, in these naturalistic experiments, the choice of listeners to approach the caller may have been influenced not only by hearing the calls of the snake detector, but also by following others who had already decided to join the caller. Investigating individual responses to the call combination and its constituent parts in the absence of other cues and behaviors from others is required to address these issues. We thus conducted playback experiments, comparing the response to the two singly-occurring calls (AH and WB) with an artificially constructed "*alarm-huu + waa-bark*" combination.

## Playback experiments

We performed playback experiments with lone individuals following a within-subjects design to control for individual reactivity to conspecific calls (N$_{trials}$ = 15, N$_{AH-WB}$ = 6, N$_{AH}$ = 5, N$_{WB}$ = 4; see below for individual distribution). GLMMs highlighted that the looking duration and latency to look towards the loudspeaker differed according to the playback condition (GLMM$_{looking-duration}$: $\chi^2 = 17.259$, Df = 2, $P < 0.001$; GLMM$_{latency-to-look}$: $\chi^2 = 33.535$, Df = 2, $P < 0.001$ respectively). Specifically, the looking duration towards the loudspeaker was significantly longer for the combination compared to both singly-occurring calls (see Table 1, Fig. 2a). Furthermore, the latency to look at the loudspeaker was significantly shorter in the WB and AH-WB conditions compared to the AH condition, but did not differ between WB and AH-WB (see Table 1, Fig. 2b). Regarding the frequency of looking towards the loudspeaker, we found no effect of experimental condition on total number of looks (GLMM$_{number-of-looks}$: $\chi^2 = 3.153$, Df = 2, $P = 0.21$). Bayesian mixed-effects models using multiple imputations to account for uncertainty resulting from the three data points missing from our within-subjects design validated these results (we aimed to test the 6 individuals in all three conditions, but could not complete 1 AH trial, and 2 WB trials; see Methods, Tables S1–S3 and Figs. S2–S4 for more details).

Since the "*alarm-huu + waa-bark*" combination represents a longer stimulus than either of the individual comprising calls, any difference in response to the playbacks (particularly in terms of looking duration) could be simply driven by the increased salience of the combination. If this was the case, in line with previous work[22], we expected the looking duration elicited by the combination to be purely additive, i.e., equal to the sum of the looking durations to the individual calls. However, we found no support for this hypothesis and subjects instead responded in more than additive ways: the response to the combination was significantly greater than the combined response to both singly-occurring calls (one-sample Wilcoxon signed rank test: $P = 0.028$).

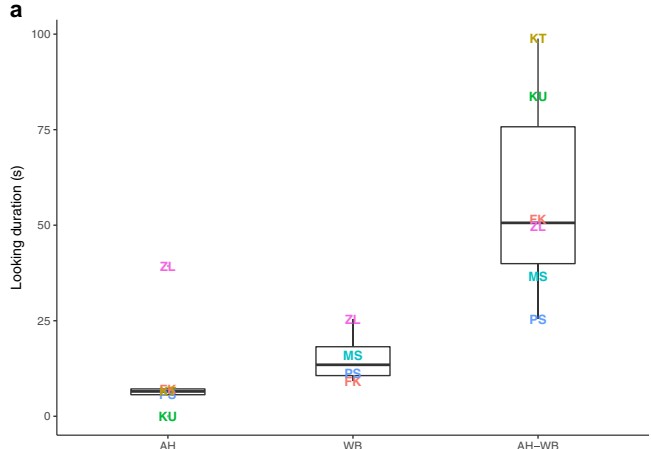

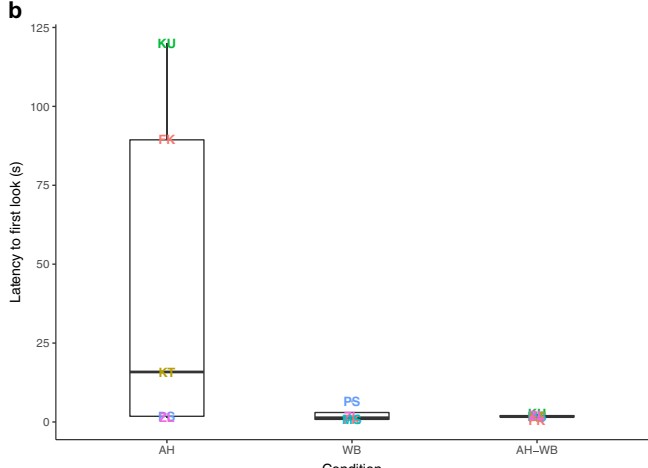

**Fig. 2 | Influence of playback condition on the looking duration and the latency to look towards the loudspeaker. a** Looking duration, GLMM$_{looking-duration}$: $\chi^2 = 17.259$, Df = 2, $P < 0.001$. **b** Latency to look, GLMM$_{latency-to-look}$: $\chi^2 = 33.535$, Df = 2, $P < 0.001$. IDs are represented with their two-letter color-coded names. AH: playback of a singly-occurring "*alarm-huu*" ($n = 5$); WB: playback of a singly-occurring "*waa-bark*" ($n = 4$); AH-WB: playback of an "*alarm-huu + waa-bark*" combination ($n = 6$). The boxes display the median value and 25 and 75% quartiles; the whiskers are extended to the most extreme value inside the 1.5-fold interquartile range. Source data are provided as a Source Data file.

Furthermore, in 3/6 (50%) AH-WB playbacks, subjects approached the loudspeaker (behavior that was not seen following any of the single call playbacks), and in two of these instances, the subject was seen engaging in typical snake anti-predator behavior, specifically climbing up a tree and looking down towards the loudspeaker (see Supplementary Movie 1). Although limited in terms of explanatory power, these anecdotal observations support the idea that the response to the "*alarm-huu + waa-bark*" sequence is more than just a combination of responses to the comprising calls.

Together, these playback results suggest the call combination is likely a concatenation of two independently occurring units which is meaningful to receivers: subjects responded differently to the combination than to single call types.

## Discussion

Understanding the evolutionary roots of the ability to combine meaning-bearing units together hinges on comparative data in our closest-living relatives: the great apes[40]. Here we provide a unique set of data combining natural observations, snake presentations and playback experiments confirming chimpanzees combine two

independently occurring calls into a larger structure and, critically, providing evidence that this call combination is meaningful to receivers.

From a production perspective, natural observations supported by snake presentations suggest the "*alarm-huu + waa-bark*" call combination is emitted in a highly specific context, namely when a chimpanzee encounters a snake, whilst the constituent calls are produced in a wide range of situations (AH: potential threat; WB: general recruitment). Thus, there is a possibility that the combination of calls may offer more specific information to listeners than the constituent calls can in isolation. From a comprehension perspective, both naturally elicited combinations and experimentally presented ones suggested receivers seem to extract a meaning from the combination since i) more individuals joined the caller after the production of the combination and ii) playback of the combination elicited the strongest responses from receivers (i.e., longest looking duration, shorter latency to look compared to "*alarm-huus*"). Together, these results suggest the "*alarm-huu + waa-bark*" functions as a recruitment signal in a dangerous situation (similarly to what has been shown in birds, see ref. 22,23).

Why should chimpanzees recruit conspecifics to a potential threat? One explanation is that snakes represent a lethal danger to chimpanzees, especially when undetected, which is very common due to their highly cryptic appearance[41]. Recruiting conspecifics in this situation might help chimpanzees encountering a snake ensure group members in the surroundings are aware of both the presence of the threat and its precise location. Indeed, a recent study has shown that chimpanzees not only extract information from alarm call production on the presence of a snake, but also infer the specific location of the snake from the physical position of the signaler who orientates their body to "mark" the snake to naive individuals[41]. This strategy seems to be very effective given chimpanzees that join an individual that has discovered a snake are not startled[42]. An alternative, non-mutually exclusive explanation is that chimpanzees recruit individuals to the threat to help drive it away, akin to mobbing behavior that has been reported in a number of species as a mechanism to deter predators or threats[43,44].

Our findings reported here are also intriguing since they bear striking resemblance with compositional syntactic structures, a core hallmark of language, where the meaning of larger phrases is derived from the meaning of the individual parts (e.g., /*danger*/ + /*come here*/). Playback experiments confirmed the meaning previously suggested for each singly-occurring call: "*alarm-huus*" elicited comparatively weak reactions from recipients (i.e., brief looks towards the loudspeaker, if at all) which was expected for potential threats[22] whilst "*waa-barks*" on the other hand elicited stronger responses (i.e., shorter latency to look at the loudspeaker) which once again was expected for recruitment calls[22], which are acoustically more attention-grabbing and often require a fast response from the receiver (to aid a victim or potentially join a hunt[37]). Finally, playback of the "*alarm-huu + waa-bark*" combination elicited the strongest responses from the subjects (i.e., longest looking duration, shorter latency compared to "*alarm-huus*"). This stronger response to the combination could be related to stimuli novelty: "*alarm-huu + waa-bark*" combinations might be more arousing simply because they are rare. This seems unlikely, however, as if this was the case, "*alarm-huus*" would elicit stronger responses than "*waa-barks*" (given the latter are much more frequently produced than the former in naturalistic communication), which is not supported by our data. Moreover, individuals seemed to respond to the combination in a more than additive way: responses to combinations exceeded the combined responses to the individual calls. These results indicate receivers did not simply respond to each call independently, but rather they seemed to extract a specific meaning from the call combination (/*recruitment to a threat*/) derived through combining the information encoded in both individual calls (AH: potential threat; WB: recruitment).

It is important to note that the data presented here do not allow us to disentangle which precise form of compositional syntax the "*alarm-huu + waa-bark*" structure most resembles. One hypothesis is that this structure represents a form of *modification* with the "*waa-bark*" directly modifying the level of urgency of the "*alarm-huu*" (i.e., from /*threat*/ to /!*THREAT*!/), akin to how the "*-oo*" affix modulates the stem alarm calls in Campbell's monkeys from labelling a specific urgent threat to communicating a more general disturbance[8,16]. Alternatively, the "*alarm-huu + waa-bark*" structure could be considered a more simple *coordination* whereby listeners bind the meaning of the comprising calls (/*threat*/; /*come here*/), to arrive at a third different but related meaning (/*threat*/ AND /*come here*/). Indeed, this interpretation has been proposed for very similar findings in birds, for instance in Japanese tits and pied babblers, who also combine a recruitment call with an alarm call which elicits a stronger response in recipients than either of the constituent calls[3,22,23]. Further experimental work could help tease the hypotheses apart. For example, if the meaning of the combination is conserved following reversal experiments, where the naturally occurring call order is switched, a coordination analysis might be favored since in this case /*threat*/ + /*come here*/ and /*come here*/ + /*threat*/ would be communicating qualitatively similar information.

In conclusion, our findings demonstrating the use and response to a call combination in our closest-living relative, the chimpanzee, suggest the foundations of syntax may be evolutionarily ancient and present in more simple forms in the last common ancestor of chimpanzees and humans. Furthermore, in conjunction with studies in monkeys, this work tentatively indicates the evolutionary origins of syntax might be traceable even further back, as far as 45 million years ago when humans and monkeys last shared a common ancestor[14]. Additional experimental work in other great ape species, especially in the *Pan* genus, will help reconstruct the evolutionary roots of syntax with an even greater degree of precision.

## Methods

### Ethical note

Ethical permission to conduct the study was received from the Animal Welfare & Ethical Review Body from the University of Warwick, United Kingdom (permit number: AWERB.35/18-19 and AWERB.01/19-20) and was further approved by the Uganda Wildlife Authority (permit number: COD/96/05) and the Uganda National Council for Science and Technology (permit number: NS47ES), Uganda.

To reduce to a minimum the impact on animals, we applied the three Rs principle. Therefore, to investigate the production of the "*alarm-hoo + waa-bark*" call combination, we analyzed snake presentation experiments conducted in 2010–11 for another study within the Sonso community (see ref. 38). Additionally, when possible, we extracted calls from the aforementioned snake presentations to generate the stimuli used for playback experiments. However, to accrue a satisfactory number of stimuli, we also conducted additional snake presentations. Finally, we conducted playback experiments on six adult individuals only – i.e. minimum sample size to detect a significant effect, and reduced the number of conditions to the bare minimum (i.e. three) to investigate whether receivers perceived the "*alarm-hoo + waa-bark*" call combination as a syntactic-like structure.

### Subject details

**Study site.** The study was conducted with the Sonso community of chimpanzees at the Budongo Conservation Field Station (BCFS), Budongo forest, Uganda (see Table S4 for the list of individuals). This community has been habituated to humans' presence since 1990[45].

### Study subjects

**Snake presentations.** Snake presentations were conducted between January 2010 and December 2011. In total, 13 individuals were exposed

**Table 2 | Details for the trials conducted with each subject**

| | | Condition | | |
|---|---|---|---|---|
| | | AH | WB | AH-WB |
| Subjects | KU (female) | X | | X |
| | FK (male) | X | X | X |
| | KT (male) | X | | X |
| | MS (male) | | X | X |
| | PS (male) | X | X | X |
| | ZL (male) | X | X | X |

*AH* singly-occurring "*alarm-huu*", *WB* Singly-occurring "*waa-bark*", *AH-WB* Artificial "*alarm-huu + waa-bark*" call combination. *KU* Kutu, *FK* Frank, *KT* Kato, *MS* Musa, *PS* Pascal, *ZL* Zalu.

to the snake in different spatial constellations, totaling 27 trials (eight males and five females, see ref. 38 for more details).

**Playback experiments**. Playback experiments were conducted between February 2019 and March 2022 for a total of 14 months. Six well-habituated adult individuals that could be routinely followed when alone were tested as subjects in playback experiments (one female: KU and five males: FK, KT, MS, PS, ZL). In total, we conducted 15 trials on 6 different individuals (see Table 2).

## Method details

### Stimuli

**Snake presentations**. The snake model used was manufactured from the skin of a dead python (*Python sebae*) donated by the Uganda Wildlife Education center (see Fig. S5). A fishing line was attached to the head of the snake to allow us to move the model from a distance (see ref. 38 for more details).

**Playback experiments**. The study consisted of three playback conditions. To determine the typical response to the comprising singly-occurring calls, we played back a single "*alarm-huu*" and a single "*waa-bark*". To investigate the function of the combination and whether its meaning is derived from the meaning of its component calls, we played back the combination of "*alarm-huus*" with "*waa-barks*". However, to ensure this call combination was truly a concatenation of the two calls produced independently (as opposed to a third, independent signal), whilst simultaneously minimizing the number of conditions chimpanzees were exposed to, we played back an artificially-constructed combination rather than a natural sequence (i.e. single "*alarm-huu*" + single "*waa-bark*"). Calls used as playback stimuli were recorded during snake presentations in 2010–11 using a Sennheiser ME67 microphone and Marantz PCM661 solid state recorder (sample rate 44.1 kHz, resolution 32 bits,.wav format). However, to accrue a satisfactory number of playback stimuli, we also conducted additional snake presentations (3D printed, hand-painted snake resembling a Gaboon viper (*Bitis gabonica*), see Fig. S6) in 2019 and recorded vocal responses using a Sennheiser ME66/K6 microphone and Marantz PMD661 mk3 solid state recorder. For the creation of playback stimuli, and to avoid introducing a sex bias, we selected high signal-to-noise ratio vocalizations from males only (MS, PS, ZL). Playback stimuli were created and normalized (in.wav format) using Adobe Audition (2015) by applying the Root Mean Square criterium[46,47]. We constructed the artificial combination by synthetically combining an "*alarm-hoo*" with a "*waa-bark*", both produced independently and originating from the same individual, with a silence of 1 s between the two calls to match naturally occurring "*alarm-hoo + waa-bark*" combinations (mean ± S.E. = 1.06 ± 0.05 s). Furthermore, to ensure the calls sounded realistic, we faded the background noise in and out to avoid the playback from having an abrupt onset and offset respectively.

In line with previous work, to ensure playbacks sounded realistic to the subject and could not be heard by the individual whose calls were broadcasted (call provider) or other individuals, we sound-tested all stimuli with a team of three experienced observers who confirmed the playback quality every 10 m from 0 to 50 m, and every 50 m from 50 to 400 m[46,47]. This allowed us to determine the optimal volume for which the playback sounded realistic to the subject while being inaudible by other individuals, primarily the call provider (at a minimum distance of 300 m). During playback experiments, the call provider was located on average 1000 m (range 500–1600 m) from the subject and was never observed to react to the playback (e.g., change of behavior, looking/orient/move towards the loudspeaker).

### Experimental design

**Snake presentations**. When a chimpanzee started travelling, one experimenter (E1) followed the individual while another experimenter (E2) moved ahead of this individual and hid the snake model under leaves and branches on its anticipated path. E2 then moved away from this location and concealed themselves from the arriving chimpanzee. When the subject approached the location of the snake model, E2 pulled the string attached to the snake model causing its immediate detection by the subject. E1 then monitored the reaction of the subject. See ref. 38 for more details on the procedure.

**Playback experiments**. All playback stimuli were played back in.wav format using a SanDisk Clip Sport Plus (www.westerndigital.com) connected to a Mipro MA-707 loudspeaker (www.mipro.com). To avoid habituation effects, the loudspeaker was concealed in a backpack (with only a hole cut into the fabric at the sound output level to avoid sound distortion) so the chimpanzees never saw the equipment in operation. Video recordings of trials were conducted using a Sony Handycam HDR-CX240 or Panasonic HC-V777 video cameras.

The protocol implemented in this study followed a similar methodology validated and used in previous work conducted in the same community of chimpanzees[46,47]. Specifically, three people were required to run the experiment to ensure aforementioned ethical and realism criteria were met. Specifically, B.C. stayed with the subject who received the playback, M.L. or C.W. played back the stimulus from the loudspeaker at a distance of ~30 m from the subject, and a third person (Denis Lomoro or Jackson Asua) followed the call provider. Contact between the three experimenters was maintained throughout the duration of the experiment using Motorola GP340 radios (www.motorola.com) and Nokia 105 phones (www.nokia.com). Subjects were initially chosen opportunistically, largely determined by which individuals were seen to isolate themselves. Trials across individuals were counterbalanced, ensuring subjects did not hear conditions in the same order, thus controlling for order effects. To avoid overexposure for chimpanzees and limit the risk of habituation, we waited a minimum of one day in between two trials from different subjects (average 27 days, range 1–104 days) and at least a week in between two trials with the same subject (average 90 days, range 28–244 days). Furthermore, we ensured that over the whole study period, no chimpanzee heard the same stimulus twice. Therefore, for a given subject, the call provider in the singly-occurring call conditions (AH and WB) was necessarily distinct from the call provider in the combination condition (AH-WB). Playbacks occurred between 0700 and 1600 h, typically around 1100 h.

We considered starting an experiment when (i) the subject was observed resting on the ground (to avoid potential bias due to the behavior/position prior to the experiment), (ii) the call provider was at least 300 m away from the subject (to ensure it did not hear its own call being broadcasted) and (iii) no other individuals were present within a 300 m radius (to avoid potential audience effect bias and ensure potential future subjects did not hear the stimulus twice). If these conditions were met, B.C. started filming the subject while M.L. or C.W.

took an indirect route to a location of ~30 m from the subject in the direction of the call provider (established through maintaining contact with D.L. or J.A.). The loudspeaker was placed out of sight of the focal subject and oriented towards it. We played the stimuli only when the subject (i) rested for at least 2 min, (ii) was clearly visible to B.C., and (iii) did not face towards the direction of the loudspeaker, so we could detect any looks oriented towards the loudspeaker. Due to the highly demanding procedure, several trials started without reaching completion and broadcasting of the stimuli, hence representing "mock" experiments allowing us to further reduce the risk of habituation. When all these additional conditions were met, M.L. or C.W. played the stimulus and B.C. video-recorded and commented the reaction from the subject, focusing specifically on the looking behavior of the chimpanzee. Finally, once a playback was completed, B.C. and M.L. or C.W. followed the subject for the rest of the day to monitor any mid- to long-term effects on the subject. No indications of stress or fear were observed from the subjects after the experiments or upon reunion with other individuals, including the call provider.

## Statistical analysis
### Data processing
**Snake presentations.** Video footage was analyzed using Boris[48]. For each trial, we noted whether the focal individual produced an "*alarm-huu + waa-bark*" combination. In line with previous work in great apes, a maximum time interval of ≤ 2 s between two calls was used to define a call combination[25,26,31,32]. We then recorded the number of individuals recruited in the two minutes after the snake was discovered. An individual was "recruited" when they came in the direct vicinity of the caller post-snake discovery (< 15 m). We excluded all trials for which we could not confidently document the aforementioned variables, resulting in the analysis of 21/27 trials.

**Playback experiments.** Video recordings were analyzed frame-by-frame using Boris[48]. Based on previous work[46,47], we noted for each trial the following behaviors from the subject:

(1) The number of looks in the direction of the loudspeaker in the 2 min after the playback, with looking direction determined by head orientation.
(2) The looking duration towards the loudspeaker in the 2 min after the playback.
(3) The latency of the first look towards the loudspeaker after the start of the playback.

Furthermore, to ensure the coding of the videos was reliable, C.W. blind coded a randomly chosen sample of 5 trials (33%), which were also coded by M.L. The mean intra-class correlation coefficient for the three variables outlined above was found to be 1.00, indicating excellent levels of coder agreement[49].

**Statistical analyses.** All statistical analyses were implemented in R. GLMMs were conducted using the lme4 and glmmTMB packages[50]. We checked model assumptions using the DHARMa package[51].

**Snake presentations.** A generalized linear mixed-effects model (GLMM) was used to investigate the influence of the production of "*alarm-huu + waa-bark*" combinations (1/0) on the number of individuals recruited at the snake (GLMM$_{individuals-recruited}$, poisson family). To control for repeated measures at the individual level and for the effect of experimental condition set in the initial study from which these data were extracted (i.e., alone, back, front; see ref. 38 for more details), we fitted ID and condition as random factors respectively.

**Playback experiments.** To investigate differences between playback conditions, we implemented a GLMM for each variable, with playback condition as the explanatory variable (AH/WB/AH-WB) and as a response variable: (i) the number of looks (GLMM$_{number-of-looks}$, negative-binomial family), (ii) looking duration (GLMM$_{looking-duration}$, gamma family) and (iii) latency to look (GLMM$_{latency-to-look}$, gamma family) respectively. To control for the independence of the variables tested, we checked for multicollinearity using the VIF function (all VIFs = -1, indicating independence[52]). Furthermore, to control for repeated measures at the individual level, we fitted ID as a random effect for each model. When a model was statistically significant, we performed post hoc analyses using Tukey tests to investigate differences between conditions. Finally, we ran Bayesian analyses using multiple imputations as an alternative from a frequentist approach to account for the small sample size and the unbalanced data set resulting from the three missing data points (MS$_{AH}$, KU$_{WB}$ and KT$_{WB}$). Indeed, the missing data may bias our conclusions if they are not random. For example, if the individuals that were not tested tend to have very strong or weak responses to playbacks generally, we may find treatment effects simply because strong/weak reactors were absent in certain conditions. To investigate whether the results of the frequentist mixed models were robust against such biases, we used multiple imputation combined with Bayesian modelling. This approach not only allowed us to account for the missing data, but also the uncertainty induced by missingness. To this end, we created 50 complete / balanced data sets using multiple mean matching to impute the 3 missing data points (using the mice function of the mice package in R[53]). We then fitted a model using Bayesian inference to each of the 50 complete data sets (using the same model structure as in the frequentist analyses), and pooled results across all 50 sub-models by combining posterior draws from all sub-models using the brm_multiple function in the brms package in R[54,55]. Each sub-model was run with four independent Markov chains of 10,000 iterations, discarding the first 5000 as a warm-up for a total of 20,000 draws per sub-model and 1 million draws across the 50 sub-models. We used default weakly informative priors for the model parameters. Reported point estimates and 95% credible intervals were calculated based on the posteriors draws across all 50 sub-models (thus reflecting uncertainty in the missing data). All Bayesian models confirmed the results from the frequentist approach (specifically biological interpretations can be offered when credibility intervals did not include zero, see Tables S1–S3 and Figs. S2–S4).

**GLMMs diagnostics.** We checked models' assumptions using the DHARMa package in R[51]. Models were not over-dispersed (GLMM$_{individuals-recruited}$: $P = 0.69$; GLMM$_{number-of-looks}$: $P = 0.76$; GLMM$_{looking-duration}$: $P = 0.53$; GLMM$_{latency-to-look}$: $P = 0.4$), no outliers were detected (all GLMMs: $P = 1$), and visual inspection of the Q-Q plots confirmed the normality of the residuals (Kolmogorov-Smirnov test, GLMM$_{individuals-recruited}$: $P = 0.45$; GLMM$_{number-of-looks}$: $P = 0.93$; GLMM$_{looking-duration}$: $P = 0.76$; GLMM$_{latency-to-look}$: $P = 0.78$).

## Reporting summary
Further information on research design is available in the Nature Portfolio Reporting Summary linked to this article.

## Data availability
The data that support the findings of this study are publicly available online at https://github.com/MaelLeroux/AH-WB/tree/Data. Source data are provided with this paper.

## Code availability
The code that supports the findings of this study is publicly available online at https://github.com/MaelLeroux/AH-WB/tree/Scripts.

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

## Acknowledgements

We thank Riwan Leroux, Robert Seyfarth and an anonymous reviewer for insightful comments and suggestions that greatly improved the manuscript. We thank UWA, UNCST and the President's office for permission to conduct the study, the BCFS staff for their constant support and the Royal Zoological Society of Scotland (RZSS) for providing core funding to BCFS. We are especially grateful to the field assistants: Geresomu Muhumuza, Monday Mbotella, Jackson Asua, Sam Adue and Denis Lomoro; without whom we could not have collected the data. We also thank Alice Bouchard, Adrian Soldati, Matthew Henderson, Marion De Vevey and Lotus Emam for their invaluable collaboration in the field that greatly facilitated experimentations, Andri Manser and the LiRI technology platform for statistical assistance, and Carel van Schaik and Balthasar Bickel for discussions. This work was supported by the Swiss National Science Foundation (PP00P3_163850 & PP00P3_198912) to S.W.T. and the NCCR Evolving Language (SNSF Agreement #51NF40_180888).

## Author contributions

Conceptualization and Methodology: M.L., K.Z., K.S., and S.W.T. Data curation: M.L., A.S., C.W., and B.C. Formal analysis: M.L. Funding acquisition: S.W.T. Resources: K.Z. and S.W.T. Supervision: S.W.T. Visualization: M.L. Writing – original draft: M.L. and S.W.T. Writing – review and editing: M.L., A.S., C.W., B.C., K.Z., K.S., and S.W.T.

## Competing interests

The authors declare no competing interests.
