## [Peer Review File · Nature Communications]

Call combinations and compositional processing in wild chimpanzeesReviewers' Comments:

Reviewer #1:

Remarks to the Author:

General: The authors present results of field observations and experiments designed to test the hypothesis that wild chimpanzees create call combinations and that these combinations exhibit the kind of 'compositional syntactic structure' that we might expect to find if this linguistic structure evolved before the divergence of humans and chimpanzees from their common ancestor. Evidence in support of this view would be extremely important; however, I don't think this paper makes a convincing case.

Line 52 Don't you mean ~25 my ago (divergence of Old World monkeys from the common ancestor)?

Line 53 can evolve through convergent evolution, complicating an account of the evolution of syntax.

Lines 54-55 But evolution within the primate lineage and convergent evolution in other taxa (like birds) are not mutually exclusive: both could have occurred. Therefore, data from great apes won't "disentangle" these explanations, because even if they supported an 'up through the primates' explanation of syntactic evolution this would not disprove an explanation based on convergent evolution in birds. The logic here needs to be re-written.

Line 98 'by the production of a call combination...' Production by which animal? The caller or one of the individuals who joined?

Line 100 How do you tell whether individuals joined because they heard the combination or because they saw other individuals joining?

Line 101 Given the data presented thus far, how do you distinguish between an explanation that posits "animals respond to the call combination as if they combine two meanings, 'alarm' and 'urgent'" and an explanation that posits "the waa-bark adds urgency to the information conveyed by the alarm-huu, much like certain acoustic changes add urgency to the alarm calls of meerkats (Manser et al. 2002, TICS), or an increase in amplitude adds urgency to speech"? The former explanation (given here) assumes a syntactic relation between the signals whereas the latter explanation does not.

Line 106 Table 1 lists only one predictor variable, whereas the text implies that several predictors were compared. The Table seems incomplete.

Line 94, Fig 1 The reader really needs to see more data. First, in the majority of presentations (12/21) no call combination was produced, so what WAS produced? Were there any presentations that elicited only an alarm-huu or only a waa-bark? If so, how strong was the response in each case? We need to see these data to determine whether the waa-bark on its own was simply a more arousing signal than the alarm-huu on its own. If this were the case, then the strong response to the call combination could have occurred simply because of the presence of the waa-bark, not the combination.

Since you go on to test these alternative hypotheses in the playback experiments, why not set out the limitations of observational data here, present the alternative possible explanations, and use the observational data to introduce the experiments? Instead, as it reads now the paper seems to imply that observational data are sufficient to prove compositionality – but they aren't.

Line 122, Fig 2 The statistics here assume that latency to look and looking duration are independent measures, but they probably are not (and they're not treated as independent in most playback experiments). Why not test them to see if they are correlated? If they aren't, then say so to justify treating them as independent. If they are correlated, you can either drop one or combine them into a single response measure using Factor Analysis/Principle Components (this is often done in song playback experiments with birds).

Line 130 I don't understand these missing data points. If there were 5 AH trials (line 119) and 1 missing AH trial (line 130), does this mean there were 4 AH trials in total? And if there were 4 WB trials (line 120) and 2 missing WB trials (line 131), does this mean there were 2 WB trials in total? The sample sizes are very small.

Line 149 Here again, how do you test between the hypothesis that adding a waa-bark to an alarm-huu simply augments the strength of response to the alarm-huu (no syntax required) versus the hypothesis that adding a waa-bark changes the meaning of the alarm-huu from what it would otherwise have been (a more syntactic explanation)? These alternative interpretations really need to be spelled out and disambiguated in detail. The paper makes two brief attempts to do so.

First, the authors show that the call combination elicits a stronger response than would be expected from the summed responses to the two calls presented independently. But this does not rule out the first explanation (augmenting the strength of response) in favor of the second (changing the meaning). Without a more detailed discussion of how the data force us to choose between these hypotheses, the case for something as important as 'compositional syntactic structure' is not convincing.

Second, the authors state that chimpanzees produce alarm-huus when surprised by a variety of stimuli – so alarm-huus, by themselves, have a comparatively vague meaning. Listeners who hear an alarm-huu know that something has aroused the caller, but don't know exactly what. However, when confronted with a specifically dangerous stimulus (often a snake), chimpanzees combine alarm-huus with waa-barks, another call with a relatively general meaning when it occurs on its own. Because this combination is given only to a restricted set of stimuli, the listener who hears it acquires new, more specific information. Thus, the call combination is compositional. Frustratingly, however, the authors never really come out and state this argument explicitly: the reader is left to infer it.

More important, the data supporting this interpretation are anecdotal. The authors' suggestion is that, upon hearing an alarm-huu, a listener has only a general expectation of what it would see if it approached the caller; upon hearing an alarm-huu+waa-bark, however, the listener's expectation is more specific. This might be true (and it's an argument that Zuberbuhler made in his habituation/dishabituation experiments), and if true it would support the hypothesis that combining the calls changes their meaning. But in the present paper it rests only on the anecdotal observation that chimps arriving after hearing the combination sometimes look for a snake.

Robert Seyfarth

Reviewer #2:

Remarks to the Author:

I think this is a very good manuscript, offering new evidence for meaningful combination of calls in a non-human primate. The data patterns are clearly presented, and, to my mind, offer solid evidence for meaningful combination of calls. This will add to the existing literature (properly reviewed in the introduction by the authors) on the evolution of 'compositionality'. The authors will no doubt be aware that specialists focusing on human language will take issue with the appeal to compositionality (some of the authors of this paper have written extensively, and usefully on this topic), but I do not think this requires more treatment than what the authors provide in the current draft.

In sum, I support publication of these results.

REVIEWER COMMENTS

Reviewer #1 (Remarks to the Author):

1. General: The authors present results of field observations and experiments designed to test the hypothesis that wild chimpanzees create call combinations and that these combinations exhibit the kind of 'compositional syntactic structure' that we might expect to find if this linguistic structure evolved before the divergence of humans and chimpanzees from their common ancestor. Evidence in support of this view would be extremely important; however, I don't think this paper makes a convincing case.

We thank the reviewer for their constructive comments and have addressed each point below.

2. Line 52 Don't you mean ~25 my ago (divergence of Old World monkeys from the common ancestor)?

Apologies for the confusion. Here we are referring to the divergence of New World monkeys from the common ancestor with humans, since combinatorial abilities have been demonstrated in both Old and New World monkeys. As far as we are aware, this split has been estimated to have occurred around 45mya. We have now modified the MS to make this clearer (see L59).

3. Line 53 can evolve through convergent evolution, complicating an account of the evolution of syntax.

This has been changed (see L71).

4. Lines 54-55 But evolution within the primate lineage and convergent evolution in other taxa (like birds) are not mutually exclusive: both could have occurred. Therefore, data from great apes won't "disentangle" these explanations, because even if they supported an 'up through the primates' explanation of syntactic evolution this would not disprove an explanation based on convergent evolution in birds. The logic here needs to be re-written.

Thank you for this point. We agree the phrasing is ambiguous. The point we wanted to make was that data from great apes will help disentangle whether combinatoriality present in humans and monkeys is a product of convergent evolution or whether it is inherited through shared ancestry. We fully agree that demonstrating similar abilities in great apes does in no way preclude a convergent evolutionary scenario in more distantly related species. We have modified the logic here to make this clearer. Please see L77-79.

5. Line 98 'by the production of a call combination...' Production by which animal? The caller or one of the individuals who joined?

By the caller. We have now modified this to make this clearer. See L132.

6. Line 100 How do you tell whether individuals joined because they heard the combination or because they saw other individuals joining?

We thank the reviewer for this point. Given the dense nature of the forest, unfortunately we could not visually track precisely what the joining individuals had seen and we agree individual decisions to join may have been influenced both by what they heard and what others did. We have modified the MS to acknowledge this possibility and to further justify the need for the playback study (see L141-151).

7. **Line 101** Given the data presented thus far, how do you distinguish between an explanation that posits “animals respond to the call combination as if they combine two meanings, ‘alarm’ and ‘urgent’” and an explanation that posits “the waa-bark adds urgency to the information conveyed by the alarm-huu, much like certain acoustic changes add urgency to the alarm calls of meerkats (Manser et al. 2002, TICS), or an increase in amplitude adds urgency to speech”? The former explanation (given here) assumes a syntactic relation between the signals whereas the latter explanation does not.

In line with the reviewer’s other comments, we have now toned down our interpretation of the production data since, as the reviewer correctly points out, these data alone are not sufficient to make any firm conclusions regarding the precise relation between the calls (i.e. syntactic or non-syntactic) (see L134-135, L136, L141-151). We now address the alternative explanations suggested by the reviewer (here and in more detail in point 13 below), when we discuss the experimental data (see our detailed response to point 13 below).

8. Line 106 Table 1 lists only one predictor variable, whereas the text implies that several predictors were compared. The Table seems incomplete.

We confirm the GLMM comprised a single predictor variable. We have therefore now removed this information from the table and integrated it into the text directly (see L133-134).

9. Line 94, Fig 1 The reader really needs to see more data. First, in the majority of presentations (12/21) no call combination was produced, so what WAS produced? Were there any presentations that elicited only an alarm-huu or only a waa-bark? If so, how strong was the response in each case? We need to see these data to determine whether the waa-bark on its own was simply a more arousing signal than the alarm-huu on its own. If this were the case, then the strong response to the call combination could have occurred simply because of the presence of the waa-bark, not the combination.

We apologise for the omission of this data from the original version of the MS. We can confirm that i) in all instances when no combination was produced an alarm-huu was produced and ii) we did not observe any snake presentations where only the waa-bark

was produced. We have now modified the MS to include this information (see L126-128). However, given the absence of waa-barks produced alone in this context, unfortunately, it is not possible to test differences in response to experimentally elicited alarm huus or waa-barks. We have now also explicitly addressed this limitation and, in line with the reviewer's below point, used it to better introduce and justify the playback experiments (see L141-151).

10. Since you go on to test these alternative hypotheses in the playback experiments, why not set out the limitations of observational data here, present the alternative possible explanations, and use the observational data to introduce the experiments? Instead, as it reads now the paper seems to imply that observational data are sufficient to prove compositionality – but they aren't.

We fully agree with the reviewer that the observational data are not sufficient to prove compositionality and agree we need to be more careful in our conclusions. We have now modified the MS to be more tentative in our conclusions and to explicitly highlight the limits of our observational data and the subsequent importance of playback experiments to resolve these limitations and disentangle alternative explanations (see L134-135, L136, L141-151).

11. Line 122, Fig 2 The statistics here assume that latency to look and looking duration are independent measures, but they probably are not (and they're not treated as independent in most playback experiments). Why not test them to see if they are correlated? If they aren't, then say so to justify treating them as independent. If they are correlated, you can either drop one or combine them into a single response measure using Factor Analysis/Principle Components (this is often done in song playback experiments with birds).

Thank you for raising this. We had previously calculated variance inflation factors (VIF) for our response variables to see if there were any issues with collinearity between them. The three response variables had low VIF values indicating they are not correlated (see table below) and suitable to be treated as independent measures.

Variables	VIF
Duration of looking towards the speaker	1.25
Number of looks to the speaker	1.22
Latency to look at the speaker for the first time	1.41

We have now added a summary of this information to the MS to clarify that the variables were not collinear (see L494-496).

12. Line 130 I don't understand these missing data points. If there were 5 AH trials (line 119) and 1 missing AH trial (line 130), does this mean there were 4 AH trials in total? And if there were 4 WB trials (line 120) and 2 missing WB trials (line 131), does this mean there were 2 WB trials in total? The sample sizes are very small.

We apologise for the confusion. We should have included information for the reader that we were implementing a within subjects design with 6 target individuals - we do this now to try and prevent confusion in the future (see L154-156 and L167-168). To clarify, we had one missing AH trial, such that 5 trials were used in the final analysis (we aimed for 6 to begin with). For the WB trials, we also aimed for 6 but missed two trials, leading to a total of 4 WB trials (see methods, table 3 L351 for detailed individual distribution).

We acknowledge the sample size is small, however the within subject design helps to increase power. In addition, as we describe in the MS, to statistically control for the unbalanced nature of the playback data and the resulting uncertainty, we ran additional imputation-based Bayesian analyses which supported our main results that playback condition had a statistically significant effect on latency to look and looking duration.

13. **Line 149** Here again, how do you test between the hypothesis that adding a waa-bark to an alarm-huu simply augments the strength of response to the alarm-huu (no syntax required) versus the hypothesis that adding a waa-bark changes the meaning of the alarm-huu from what it would otherwise have been (a more syntactic explanation)? These alternative interpretations really need to be spelled out and disambiguated in detail. The paper makes two brief attempts to do so.
- First, the authors show that the call combination elicits a stronger response than would be expected from the summed responses to the two calls presented independently. But this does not rule out the first explanation (augmenting the strength of response) in favor of the second (changing the meaning). Without a more detailed discussion of how the data force us to choose between these hypotheses, the case for something as important as 'compositional syntactic structure' is not convincing.

We thank the reviewer for this point and acknowledge that we should have supported our findings and interpretation better through more clearly outlining the various key forms of syntactic structures that can occur and which ones have already been demonstrated in non-human animals. We have now modified the introduction to address this issue (see L45-53, L60-70, L72-77). To summarise, animals can combine calls to form a combination which has a new meaning or function that is unrelated to the parts (i.e. in the putty-nosed monkey system where hack = eagle, pyow = disturbance and pyow-hack = lets move), which has been termed "combinatorial syntax" (Hurford 2011)). Call combinations can also occur where the meaning or function of the combination is directly, transparently related to the parts ("compositional syntax" Hurford 2011). Compositional syntax can be further subdivided into different types of combinations, of which two key forms are modification and coordination (Hurford,2011). Modification is where one call modifies the meaning of another (e.g. the Campbell's monkey system where one call (-oo call) modifies the urgency of the other call (Kraak or Hok)). Coordination is where the meaning

of two calls are linked together to arrive at a third, related meaning (e.g. alarm and recruitment calls combined in pied babblers and Japanese tits to denote a threat that requires recruitment, Engesser et al. 2016, Suzuki et al. 2016, Townsend et al. 2018).

It is our understanding that the “augmenting the strength of response” explanation as suggested by the reviewer would be analogous to the effect of the -oo affix on the stem alarm calls in the Campbell’s monkey (though in the Campbell’s it reduces the urgency rather than increases it) and therefore would still represent a form of compositional syntax (modification: see Collier et al. 2014, Townsend et al. 2018). The other possibility suggested by the reviewer whereby the waa bark “changes the meaning” of the alarm huu, in our opinion, would be more reminiscent of the pyow-hack combination where the addition of the pyow changes the meaning of the hack from announcing the presence of an eagle to initiating group movement. Given the contextual data seems to suggest an overlap in function or meaning of the “Alarm-huu+Waa-bark” combination and the comprising parts, we think this hypothesis is less likely and instead we focus on ‘coordination’ as an alternative form of compositional syntax that may also apply to our findings. Given the similarity between the data from birds combining alarm and recruitment calls and demonstrating ‘coordination’, and our findings, we think this is an important hypothesis to explore.

Through adding a description of these various types of syntactic “operations” in the introduction (see L45-53, L60-70, L72-77) and referring back to them in the discussion (L302-317) we have now endeavored to better clarify what we think are the most convincing hypotheses for the identified combination. We are also clear that our current data do not allow us to conclude which type of compositional syntactic structure best describes the “Alarm-huu+waa-bark” combination and we now temper our interpretation of the data (see L259, L298, L302-313) and suggest some future experimental work that may indicate whether this combination is best conceived as an example of modification or coordination (see L313-317).

Second, the authors state that chimpanzees produce alarm-huus when surprised by a variety of stimuli – so alarm-huus, by themselves, have a comparatively vague meaning. Listeners who hear an alarm-huu know that something has aroused the caller, but don’t know exactly what. However, when confronted with a specifically dangerous stimulus (often a snake), chimpanzees combine alarm-huus with waa-barks, another call with a relatively general meaning when it occurs on its own. Because this combination is given only to a restricted set of stimuli, the listener who hears it acquires new, more specific information. Thus, the call combination is compositional. Frustratingly, however, the authors never really come out and state this argument explicitly: the reader is left to infer it. More important, the data supporting this interpretation are anecdotal. The authors’ suggestion is that, upon hearing an alarm-huu, a listener has only a general expectation of what it would see if it approached the caller; upon hearing an alarm-huu+waa-bark, however, the listener’s expectation is more specific. This might be true (and it’s an argument that Zuberbuhler made in his habituation/dishabituation experiments), and if true it would support the hypothesis that combining the calls changes their

meaning. But in the present paper it rests only on the anecdotal observation that chimps arriving after hearing the combination sometimes look for a snake.

As suggested by the reviewer, we now explicitly highlight the possibility that the combination gives more specific information to listeners than the constituent calls (L254-258). However, we also agree with the reviewer that the anecdotal observations of listener responses that support this possibility are limited in their explanatory power given their infrequent occurrence and we explicitly acknowledge this on L182-185. Consequently, we no longer discuss these anecdotal data in the discussion section of the paper (see L276-278).

Lastly, we have also made some minor changes to the MS text here and there to aid readability and clarity.

Reviewer #2 (Remarks to the Author):

I think this is a very good manuscript, offering new evidence for meaningful combination of calls in a non-human primate. The data patterns are clearly presented, and, to my mind, offer solid evidence for meaningful combination of calls. This will add to the existing literature (properly reviewed in the introduction by the authors) on the evolution of 'compositionality'. The authors will no doubt be aware that specialists focusing on human language will take issue with the appeal to compositionality (some of the authors of this paper have written extensively, and usefully on this topic), but I do not think this requires more treatment than what the authors provide in the current draft.

In sum, I support publication of these results.

We thank the reviewer for their comment and their positive feedback.

Reviewers' Comments:

Reviewer #1:

Remarks to the Author:

As the authors have addressed pretty much all of my comments in their revised version, I have nothing further to suggest.